# Aluminum Bronze Crystallization on Deformed Base during Electron Beam Additive Manufacturing

Anton Y. Nikonov [1,2] , Dmitry V. Lychagin [1,2,*], Artem A. Bibko [1,2] and Olga S. Novitskaya [1]

1   ISPMS Institute of Strength Physics and Material Science SB RAS, Akademicheskii pr. 2/4,
    634055 Tomsk, Russia; anickonoff@ispms.ru (A.Y.N.); artem.bibko@mail.tsu.ru (A.A.B.);
    nos@ispms.tsc.ru (O.S.N.)
2   Department of Metal Physics (A.Y.N.), Department of Mineralogy and Geochemistry,
    TSU Tomsk State University, Lenin Av. 36, 634050 Tomsk, Russia
*   Correspondence: lychagin@mail.tsu.ru; Tel.: +7-3822-529-447

**Abstract:** To obtain products by using additive manufacturing (AM) methods, it is necessary to take into account the features of the formed internal structure of the material. The internal structure depends on the 3D printing parameters. To predict it, it is effective to use computer modeling methods. For this purpose, using the example of aluminum bronze, the influence of the base structure and heat input during surfacing on the grain structure of the deposited layers was studied. To create numerical models, we used data obtained from electron backscatter diffraction (EBSD) analysis of samples. The heterogeneity of the formation of the structure in each selected zone is established, which indicates the heterogeneity of heat input in local areas of the material in one mode of surfacing. For typical cases of crystallization, modeling using the molecular dynamics (MD) method of crystallization processes with different heat inputs to the base with characteristics specified based on experimental data was carried out. It was established that the amount of heat input determines the degree of melting and the inherited defectiveness of growing crystals. The formation of misorientation boundaries and crystallization centers of new grains is determined by the conditions of joint growth of grains with given crystallographic parameters of the computational model. The grain structure obtained as a result of simulation is consistent with the experimentally observed structure of the samples.

**Keywords:** additive manufacturing; aluminum bronze; molecular dynamics simulation; electron backscatter diffraction

## 1. Introduction

Additive manufacturing of metal materials has taken the manufacturing process of small-scale production to a new level. Using the achievements of welding production and automatic design systems, it became possible to obtain complex products that practically do not require finishing operations. Additive manufacturing, or rapid prototyping, is a significant development in the field of manufacturing processes. Extensive research in this area of 3D printing is summarized in reviews of materials, methods, applications, and challenges. The methods have their own characteristics, advantages, and limitations and are applicable to printing a certain range of materials [1]. There are reviews on aerospace [2], membrane technology [3], strategies for the optimum part build [4], and additive/subtractive hybrid manufacturing of directed energy deposition (DED) [5]. Liu S. et al. compared progress in Ti6Al4V fabricated by DED, selective laser melting (SLM), and electron beam melting (EBM) [6]. Herzog D. et al. considered the complex relationship between AM processes via SLM, EBM, and laser metal deposition, microstructure and the resulting properties for steel, aluminum, and titanium alloys [7]. Aboulkhair N.T. et al. [8] and Olakanmi E.O. et al. [9] systematized the most influential process parameters for the 3D printing of aluminum alloys using SLM. They considered design, microstructures, crystallographic texture, heat treatment, and mechanical properties. Post-processing techniques apply to the resolution

of defects in AM. Laser shock peening, laser polishing, conventional machining methods, and thermal processes are usually applied [10]. The structural design of materials and its adaptation for additive manufacture are considered in reviews [11,12]. A comprehensive overview of the physical processes and the underlying science of the metallurgical structure and properties of the deposited parts is provided.

Recently, cases of using methods that combine the parameters of several AM methods have not been uncommon. The work in [13] demonstrates the use of a combined additive printing method that combines the supply of powder and wire. Using the CuAl–WC system as an example, it was studied how the concentration and particle size of the reinforcing material affect the microstructure characteristics and tribological properties of the resulting composite material.

When surfacing using different feed rates of several wires of different chemical compositions, it is easy to automate the process of creating products with varying composition and structure or to rebuild equipment for the manufacture of products that differ in composition. The authors of the work in [14], using this technology, obtained a series of Ti-6Al-4V alloys with different copper contents and established a composition variant with the best structure and mechanical properties.

An attractive feature of additive technologies is the ability to quickly and automatically change surfacing modes. This allows the control of the structure of printed material layers, obtaining gradient structures, and creating flexible and easily reconfigurable production lines. Many works are devoted to the study of the influence of surfacing parameters on the structures and properties of printed products. Liu Y. et al. [15] investigated a Ti-22Al-25Nb alloy fabricated via SLM at different scanning speeds and found that scanning speed significantly affects the characteristics of microstructures and the separation of the secondary phase.

An attractive point of additive technologies is that when they are used, it is possible to control the grain morphology and the structure of printed materials. It is possible to obtain the required grain structure of the resulting material using variations in the surfacing mode, intermediate deformation, or heat treatment [1,4,16–19].

The generally accepted approach to improving the functional characteristics of bronzes is heat treatment [20,21]. The change in the structure of aluminum bronze can be associated with phase transformations that occur in the presence of temperature effects. Thus, it was shown in [22] that during the annealing of aluminum bronze obtained via electron beam additive manufacturing (EBAM) in the temperature range of 400 °C, 675 °C, and 800 + 400 °C, the decomposition of the $\beta'$-phase led to a decrease in the tensile strength and increasing plasticity. The paper in reference [23] describes in detail how heat input affects the formation of growth structures, and how the mechanical properties of Al bronze grown using the EBAM method depend on this parameter. In [24], it is shown how an equiaxed structure formed by static recrystallization was obtained using thermal and thermomechanical treatments for aluminum bronze samples obtained via AM. Rolling, friction stir processing, hot isostatic pressing, and shot peening are also applicable to samples obtained via wire arc additive manufacturing, selective laser melting or EBAM to influence the structure and mechanical properties [25–27]. The authors of the review in [28] described in detail the features of DED, concluding that post-processing has a positive effect in a number of cases. The most frequent courses of events under such impacts are phase transformations and recrystallizations. In [29], using a model experiment as an example, a positive effect of interlayer impact treatment on the mechanical characteristics of CuAl7 bronze obtained via EBAM is shown. In particular, the use of interlayer impact treatment promotes the formation of equiaxed recrystallized grains with annealing twin boundaries.

When developing technological processes for welding and surfacing, designing products obtained via 3D printing, and obtaining products with desired properties, modeling of production processes is promising. Full simulation is a complex, multi-level task that requires significant computing power. Dal M. and Fabbro R. distinguish two groups of models: thermomechanical and multiphysical [30]. Marques E.S.V. et al. distinguish them

as three stages: thermal modeling, metallurgical modeling, and mechanical modeling [31]. To take into account deformations during welding and surfacing, it is necessary to carry out thermomechanical modeling [24–29,32,33]. The finite element method allows the prediction of residual stresses and deformations that depend on temperature gradients [34]. Ding J. et al. [35,36] used FEM to calculate temperature, strain, and residual stress distributions for steel. Casuso M. et al. [37] developed an FEM-based model for evaluating the strains that occur during gas metal arc welding. This model predicts temperature with good accuracy and qualitatively estimates the resulting deformations of the product. To perform thermomechanical analysis of nickel aluminum bronze alloys in a laser hot-wire-directed energy deposition additive manufacturing process, Hatala G.W. and colleagues developed an FE model that takes into account multiple heat sources [38]. Thermal influence and the resulting mechanical stresses and deformations lead to structural changes. When modeling, it is also necessary to take into account the phase transformations of steel [39,40]. The transformation in the crystal structure during surfacing is modeled using the finite element method [41] and the cellular automata method [42] in comparison with the results of the EBSD analysis.

The molecular dynamics method is often used as a tool for studying on an atomic scale the mechanisms of material crystallization [32,33,39,40,43]. A detailed review of methods for modeling the characteristics of microstructures in the additive production of metals is given in [44]. Interesting results are presented by Vo T.Q. and Kim B.H. in [45] on thermal and energy management for additive manufacturing. Zhou L. et al. [46] present the results of MD simulations, which make it possible to improve understanding of the crystallization process of FCC alloys during rapid cooling. Singh G. et al. in [47] discuss the relationship between the parameters of the additive process and the crystal structure using the example of copper, and state that it is possible to avoid defects with slow cooling. Grain growth processes can be affected not only by temperature parameters but also by external influences, such as a load or an electric field [48]. The molecular dynamics method is one of the applicable methods for predicting detailed grain morphology [49]. Although MD simulation is associated with the use of large computational resources, it provides information on the structural features of the material, including helping to trace the crystallographic dependences of growth [50]. It becomes possible to trace the interaction of a melt drop with the crystallographic orientation of the substrate grains and the regularities in the formation of stacking faults and twins [51].

The use of MD simulation makes it possible to predict the most accurate structure in nanovolumes and to trace the change in the defect structure inside grains under thermal exposure. The simulation data can be compared with the results of the crystallographic analysis obtained using the EBSD method. The use of MD modeling can supplement the results obtained via the finite element method and cellular automata.

In this regard, the aim of the work is to study the grain structure of aluminum bronze deposited on a deformed base and compare it with the results of MD modeling of the crystallization process.

## 2. Materials and Methods

The objects of research are samples of aluminum bronze containing 7.5 wt.% aluminum (Cu-13 at.% Al). A work piece was obtained via 3D electron-beam building-up in a vacuum. Building-up was carried out on a laboratory installation of ISPMS SB RAS. Additive production modes are presented in Table 1. The modes used were selected on the basis of previously completed works [23]. Surfacing was carried out using a zigzag motion of the table relative to the gun in layers along programmed transitions to a new layer in a time of 30 s. The product obtained via 3D surfacing was deformed via compression by a 0.75 nominal strain. Then, the layers were deposited on the prepared surface according to similar modes. The sequence of operations is shown in Figure 1.

**Table 1.** Parameters used for 3D electron-beam building-up.

| Electron Beam Accelerating Potential (kV) | Beam Current (mA) | Spot Size (mm) | Beam Sweep Frequency (Hz) | Heat Input (kJ/mm) |
| --- | --- | --- | --- | --- |
| 30 | 30 | 4.5 | 1000 | 0.22 |

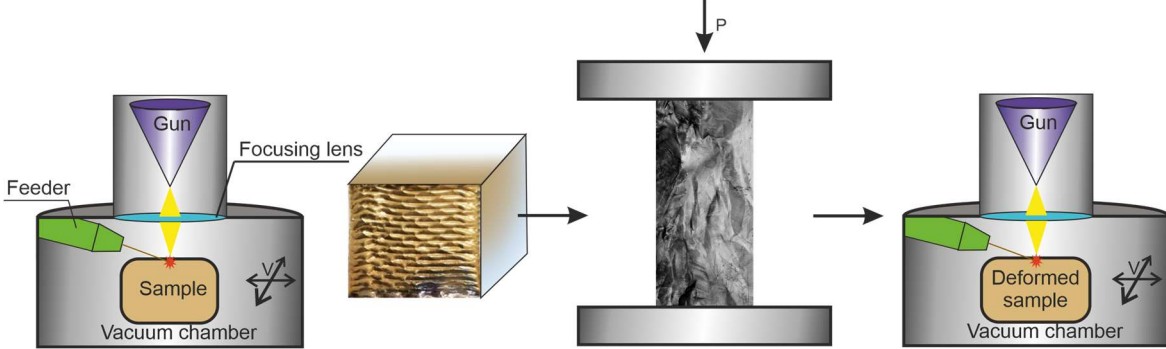

**Figure 1.** Stages of sample preparation.

The resulting workpiece had the following dimensions: a length of ~30 mm, width of ~25 mm, and height of ~18 mm. The sample cut from the workpiece had the following dimensions: a length of ~16 mm, width of ~7 mm, and height of ~6 mm. The size of the surface under investigation is $7 \times 16$ mm$^2$. The orientation of the surface is parallel to the direction of the movement of the melt spot. The long side of the sample was oriented perpendicular to the building-up layers. The sample was cut out via electrical discharge machining. The surface of the samples for the study was prepared according to a standard technique, including polishing with abrasive paper and polishing suspensions. The final stage of preparation was surface ion milling with a low-energy ion beam on a SEMPrep2 device (Technoorg Linda Co. Ltd., Budapest, Hungary). The grain orientations and grain boundary misorientations were studied using the electron backscatter diffraction (EBSD) method (Instrument Nordlys, Oxford Instruments, High Wycombe, UK) with an instrument mounted on a Tescan Mira 3 LMU scanning electron microscope (TESCAN ORSAY HOLDING, Brno, Czech Republic). HKL Channel 5 software (Oxford Instruments, High Wycombe, UK) was used for an analysis of the EBSD data [52]. EBSD investigation was carried out with the equipment of Tomsk Regional Core Shared Research Facilities Center of NR TSU. The center is supported by the Ministry of Science and Higher Education of the Russian Federation, grant no. 075-15-2021-693 (no. 13.RFC.21.0012).

## 3. EBSD Crystallographic Analysis of Grains Obtained via Electron Beam Surfacing

In this section, the results of morphological and crystallographic analyses of the aluminum bronze grain structure after surfacing on a deformed base are considered. Surfacing was carried out via the electron-beam method using aluminum bronze wire in the same modes before and after deformation.

An overview image of the transition area from the deformed part of the sample to the deposited layers is shown in Figure 2 (X is the direction perpendicular to the deposited layers from the base, located on the left). According to the structure relative to the boundary between the deformed and deposited material, we distinguish three zones: the heat-affected zone, the remelting zone, and the deposited alloy zone. The heat-affected zone highlighted in the figure extends to 7–8 mm and is characterized by areas of return and partial and complete primary recrystallization, as well as an area of secondary recrystallization. Deformed grains are observed outside the heat-affected zone.

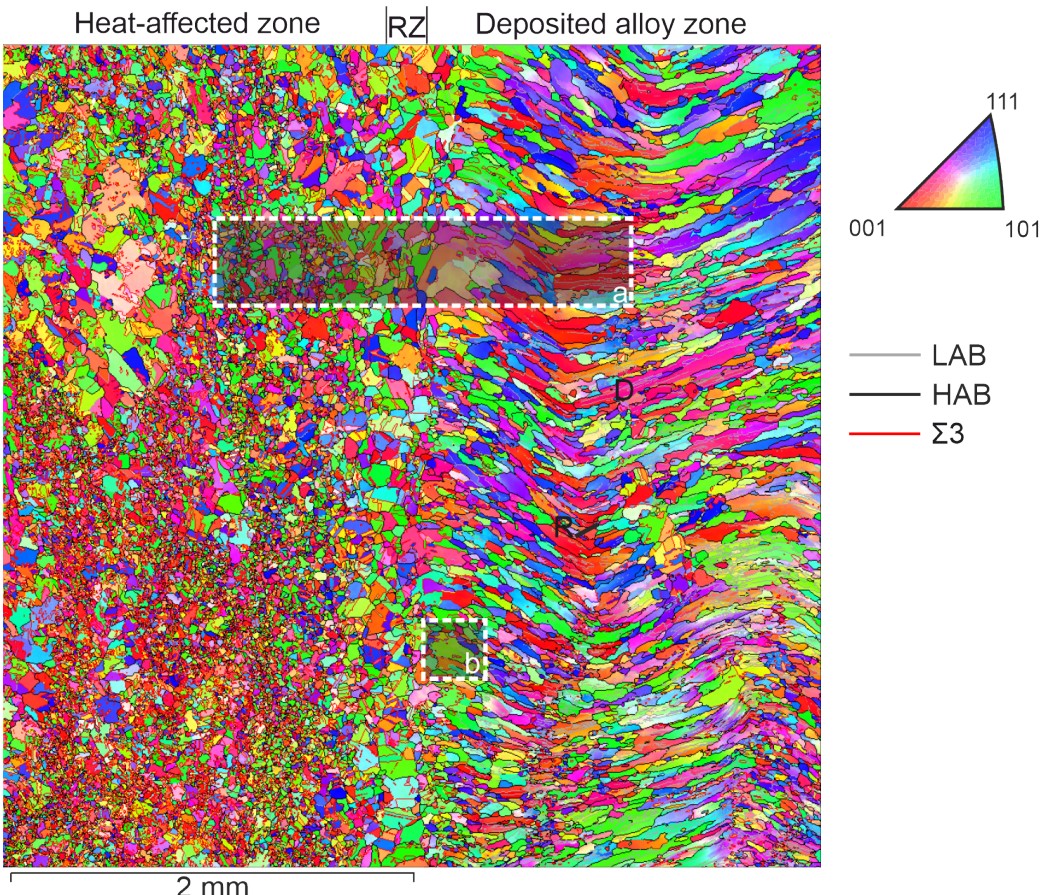

**Figure 2.** The orientation grain map relative to the X axis in IPF colors and the main types of boundaries. Sections a and b are selected zones of sample; R—fine grains, D—columnar grains.

An overview image (size 4 × 4 mm$^2$) gives us an idea of the heterogeneity of the development of the substructure in depth and in terms of the direction of surfacing. The thickness of each deposited layer is approximately 0.5–1 mm. A remelting layer separates the surfacing layers. The thickness of the remelting layer reaches 200 μm. When a new layer is deposited, we observe crystallization processes with the formation of fine grains (R on Figure 2) or the continued growth of columnar grains in the direction of a new maximum temperature gradient (D on Figure 2). Columnar grains change their slope following the thermal field during surfacing when the direction of movement of the workpiece relative to the electron gun changes during the surfacing of the next layer. The marking of the grains in the colors of the inverse polar figure indicates the predominant orientation of individual grains and groups of grains in different parts of the deposited zone relative to the chosen orientation of the laboratory coordinate axis. Fine and medium-sized grains represent the structure of the remelting zone at the boundary with the deformed material. This zone is limited on one side by a layer of secondarily recrystallized isometric grains and on the other side by deposited grains. Moreover, the first deposited layer has a finer-grained structure than the subsequent layer, with more ideal columnar grains. Conventionally, they are called unstable and stable zones.

The main difference in the structure lies in the difference in the polycrystal's grain morphology. The crystallography of grains (texture) and the characteristics of the grain-boundary ensemble associated with the misorientation angle and the crystallographic orientation of the rotation axis differ. It is possible to distinguish the proportions of low-angle (LAB) and high-angle (HAB) boundaries and special-type boundaries (STB). Figure 3 is section A in Figure 2 showing a more detailed description of the structure.

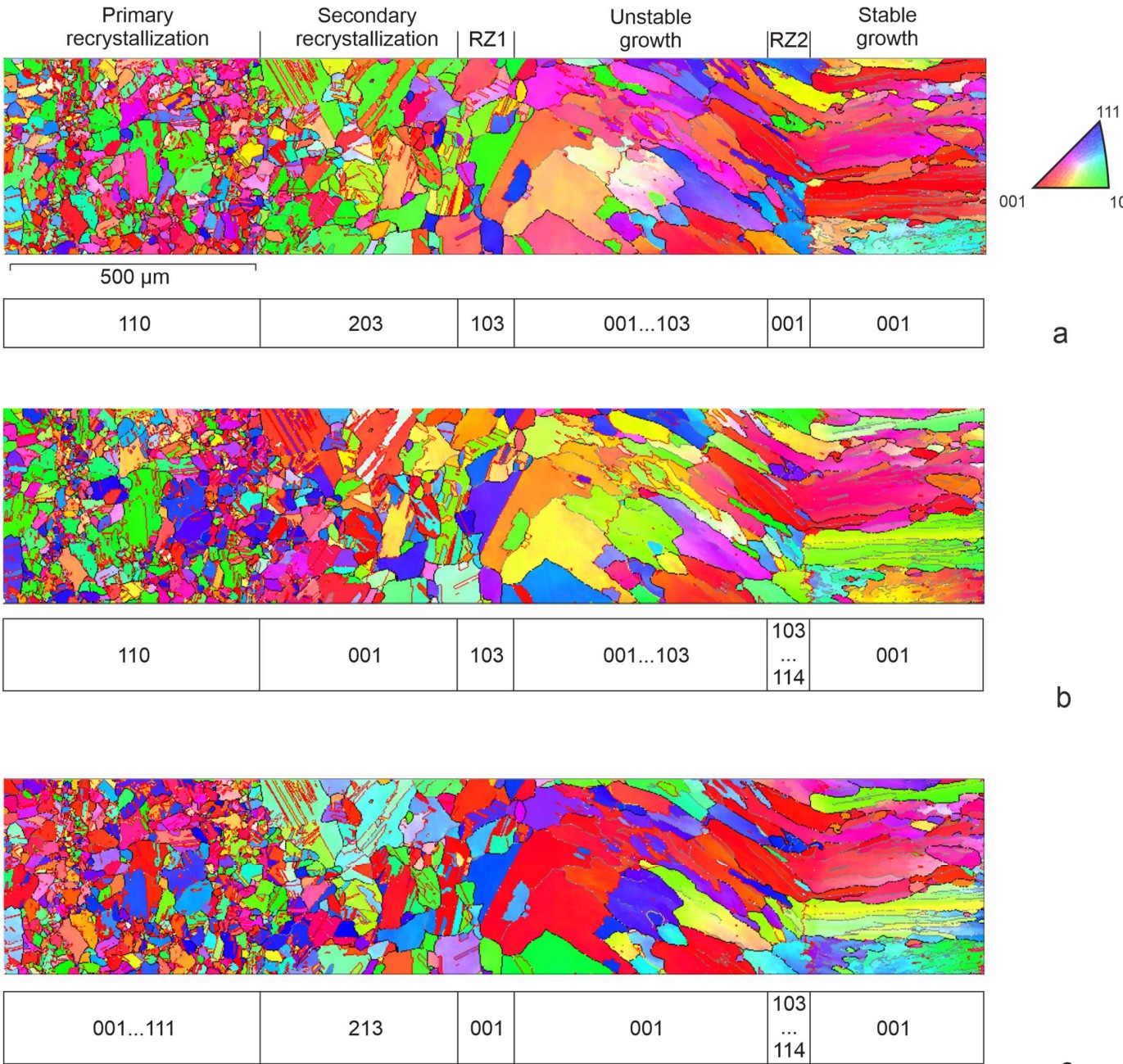

**Figure 3.** The orientation grain maps in IPF colors (**a–c**) and preferred crystallographic orientations of the zones relative to the X, Y and Z axes.

The orientation grain maps in IPF colors (Figure 3a–c) give an idea of the grain orientation related to the deposition direction (X-axis) and to the lateral faces of the bar (Y and Z axes). The predominant crystallographic orientation of grains relative to these directions was identified based on IPF. Primary recrystallization of aluminum bronze leads to the formation of numerous fine grains. First, they are formed along the boundaries of groups of deformed grains. In the zones of primary and secondary recrystallization, as well as in the zones of remelting, the preferential grain orientation is not observed. However, in the zones of the deposited area, there is a tendency of the cubic orientation to predominate. In neighboring parallel sections, the preferred orientation and sequence of orientation changes may be different (Figure 2).

The analysis of grain boundaries is of particular interest. Figure 4 gives an idea of the types of grain boundaries. The qualitative picture is illustrated by the boundary map (Figure 4a) and the change in the proportion of boundaries in the area of heat-affected zones and deposited zones (Figure 4b). Separation of LAB and HAB was carried out according to the misorientation value of 15°. The formation of low-angle boundaries is observed in the system of columnar grains of deposited layers. Apparently, this is due either to the incompatibility of the growth of neighboring grains or to thermal stresses arising in the deposited layers after surfacing. The share of HAB decreases in the deposited layers, while that of LAB increases. More than half of the HABs are of STBs. These are the Σ3 boundaries, which are 60° misorientation boundaries around the <111> axis. Their main part is the annealing twin boundaries inside the grains. Special-type boundaries Σ9 with a rotation of 38.94° around the <110> axis make up a few percent of all high-angle boundaries. A large number of STBs are formed in the heat-affected zone. Their share decreases by almost two times with a stable growth of columnar grains in the surfacing zone.

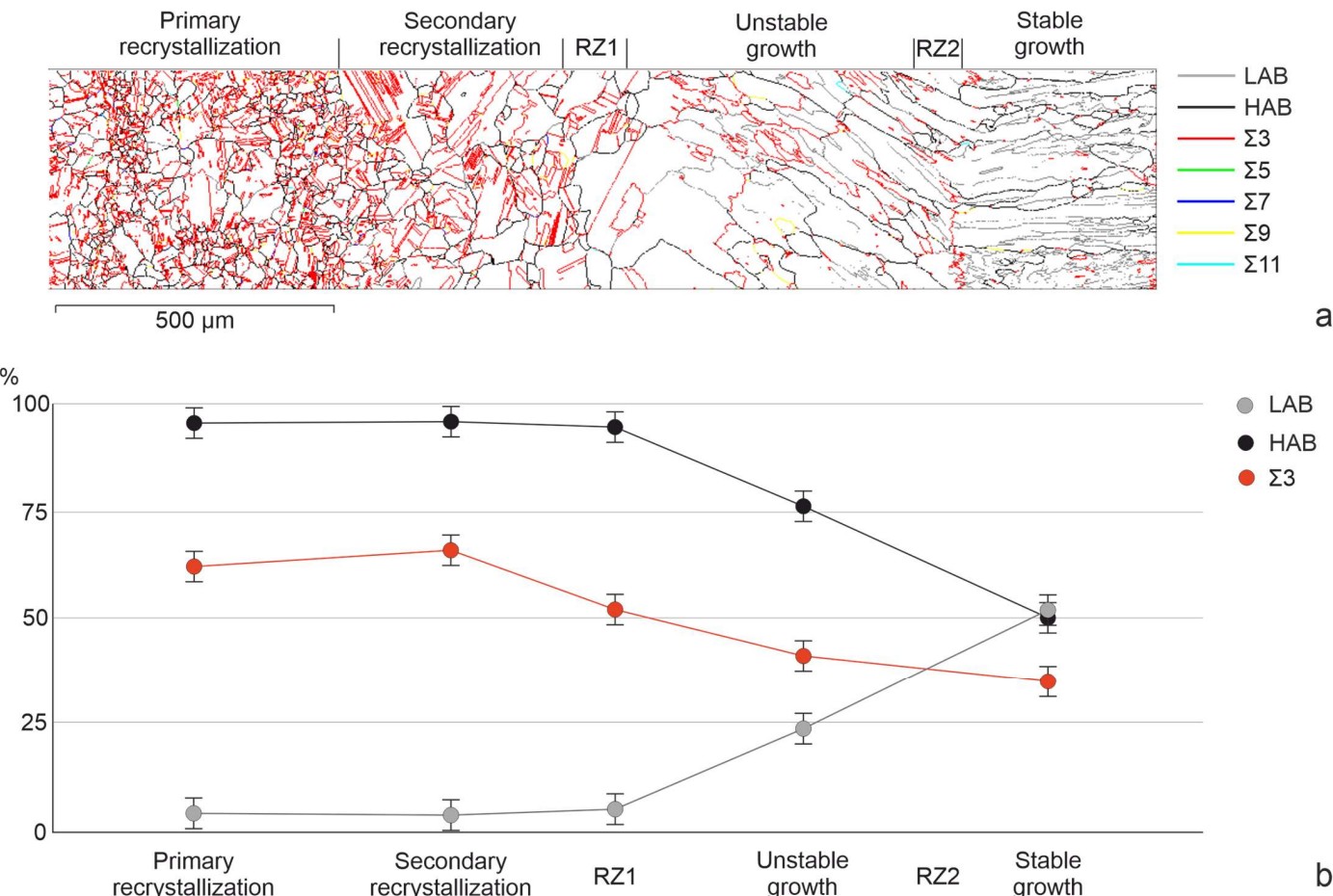

**Figure 4.** The maps of the Σ3 boundaries of LABs, HABs and STBs (**a**) and their share in the identified zones (**b**).

Thus, the experimental data provide information about the processes of thermal influence on the deformed base during surfacing and the growth of new grains. Data on crystallographic orientation and types of boundaries are the initial data for the problem of molecular dynamics simulation.

## 4. Model Description

Simulation was performed via the molecular dynamics method using the LAMMPS software package (version 29 October 2020, Sandia National Laboratories, Livermore, CA,

USA) [53]. The interatomic interaction of the selected materials was described using the interatomic potential constructed within the second-nearest-neighbor modified embedded-atom method (2NN-MEAM) that was developed for binary aluminum (Al) alloys applicable from room temperature to the melting point [54]. The equations of atomic motion were integrated using the Verlet velocity method. The OVITO software (version 3.7.11, OVITO GmbH, DE) [55] was used to visualize and analyze the faulted structure. The internal structure was studied using the polyhedral template matching (PTM) method [56]. This method allows you to determine the crystal lattice in which the atom is located according to its nearest environment and to calculate the orientation of the lattice relative to the laboratory coordinate system. Based on these data, various grains can be distinguished in a polycrystalline sample. In addition, by noting the atoms located at the nodes of the hcp lattice, one can reveal such defects as a stacking fault and a twin boundary.

The simulation objects were groups of grains from selected areas of a Cu-13 at.% Al aluminum bronze sample obtained by 3D surfacing. The crystallographic parameters of these grains were taken from the results of the EBSD analysis. It was assumed that the grain boundaries were located perpendicular to the image plane. The simulation was carried out for two cases: (a) surfacing on a base with a crystallization structure (a preliminary deposited layer) and (b) surfacing on a deformed base.

### 4.1. Surfacing on a Base with a Crystallization Structure

The deposition of the molten Cu-13 at.% Al alloy was carried out on a polycrystalline substrate that was $5 \times 22 \times 11$ nm$^3$ in size. During crystallization, two cases were considered. In the first case, a melt drop with a temperature of 1500 K interacted with the base grains, which played the role of seed elements during crystallization (grains 1, 2, and 3 on Figure 5, which is section B in Figure 2). The heat was removed only from the side of the base. In the second case, an additional crystallization center was formed inside the melt drop. To carry this out, after the contact of the substrate with the drop, a part of the drop volume ($4 \times 4 \times 5$ nm$^3$) was replaced by a crystal structure with a certain orientation (grain 4 on Figure 5b). The sample in this case was cooled from the side of the base and from the side of the new grain. The simulation of heat removal was set by setting the atoms of a 3 nm thick layer near the substrate and the atoms of the crystallization center in a drop of additional viscous forces, the value of which was calculated using the formula F = −kV, where V is the atomic velocity and k is the proportionality coefficient. To prevent the drop material from going past the sample, "virtual" walls of the simulated system were set. The walls were located at a distance of 0.5 nm from the sample surface.

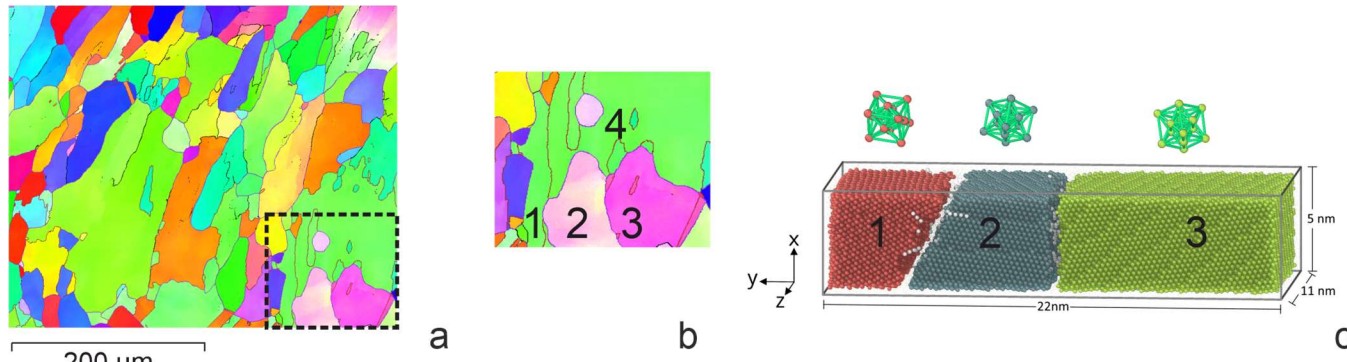

**Figure 5.** Area selected for simulation (**b**) on the orientation map (**a**) (section b in Figure 2) and the structure of the model before interaction (**c**) (1–4 are grains whose parameters were put into the model). Here and below, the colors show the atoms belonging, according to PTM, to different grains. Small dots show atoms that do not have a definite crystal structure (melt, interfaces, structural faults, etc.). Atoms located at the sites of the HCP lattice are marked in white.

### 4.2. Surfacing on a Deformed Base

The role of heat input during surfacing was simulated using the process of interaction of a molten drop with a cold base that was $12 \times 36 \times 12$ nm$^3$ in size, consisting of three deformed grains. Based on the results of the crystallographic analysis of the sample, a typical group of three deformed grains was selected, which served as the basis for setting the parameters for simulation. The parameters of the initial structure of the base were set based on the analysis of the orientation map of the sample obtained via the EBSD method. The faulted structure of grains was created via plastic deformation in a model experiment [51]. Its structure after deformation is shown in Figure 6. The interaction of the base with a drop of different temperatures was simulated near the melting temperature of aluminum bronze, 1380 K (overheating by 50 degrees), as well as at temperatures of 1500 K and 2000 K. Heat removal was carried out from the side of the base.

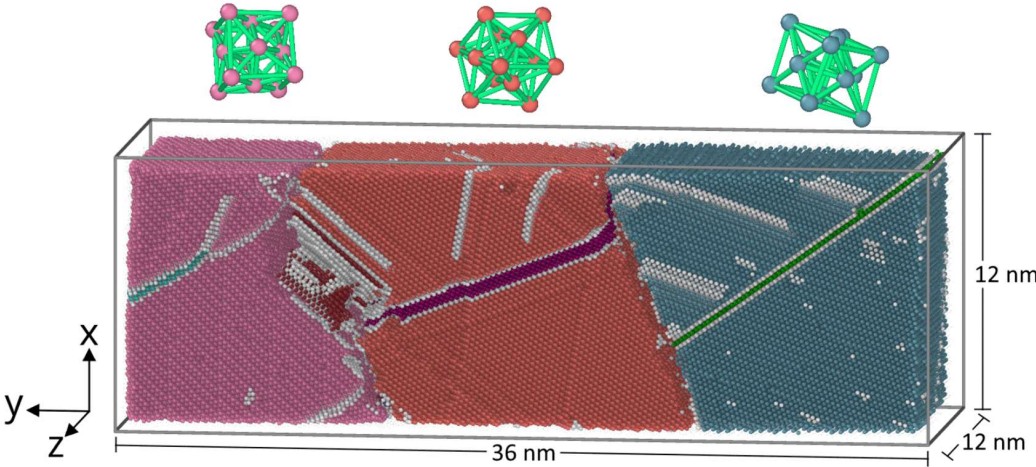

**Figure 6.** Three-grain model structure after deformation before interaction with a melt drop.

Thus, the considered cases correspond to the processes of crystallization of the first layer on the deformed base and the second and subsequent layers deposited on the crystallized layer. The sequences of structural changes in the "drop-base" system are determined by the patterns of heating and cooling of the components in this system. The indicator of this process is the temperature-related velocity of the atoms.

## 5. Results of Simulation and Comparison with Experiment

### 5.1. Surfacing on a Base with a Crystallization Structure

The structure of the sample obtained via deposition of a molten drop on a polycrystalline substrate with grain orientations of grains 1–3 (Figure 5) at the moment of maximum melting is shown in Figure 7a. Figure 7b illustrates the result of a process of simulation with an additional crystallization nucleus in a melt drop. During heat removal, the melt drop cooled down and crystallized (Figure 7c,d). In the first case, grain growth occurred due to the growth of grains 1, 2, and 3 of the base. In the second case, a grain in a drop crystallized simultaneously (grain 4). Stacking faults (SF) were formed in growing grains (SF in Figure 7c,d). Along with the main grains of crystallization, the orientation of which was set at the initial moment, twins were formed (T in Figure 7c). Satisfactory agreement with the experimental patterns of grain crystallization was observed. For example, we see the growth of the twin (T1) at the grain boundary in Figure 7c and the similar formation of the twin (T1) at the grain boundary in Figure 7e. In general, if we take into account the limited volume of simulation in MD, the qualitative patterns in the modeling of crystallization and those observed in the process of surfacing are very similar.

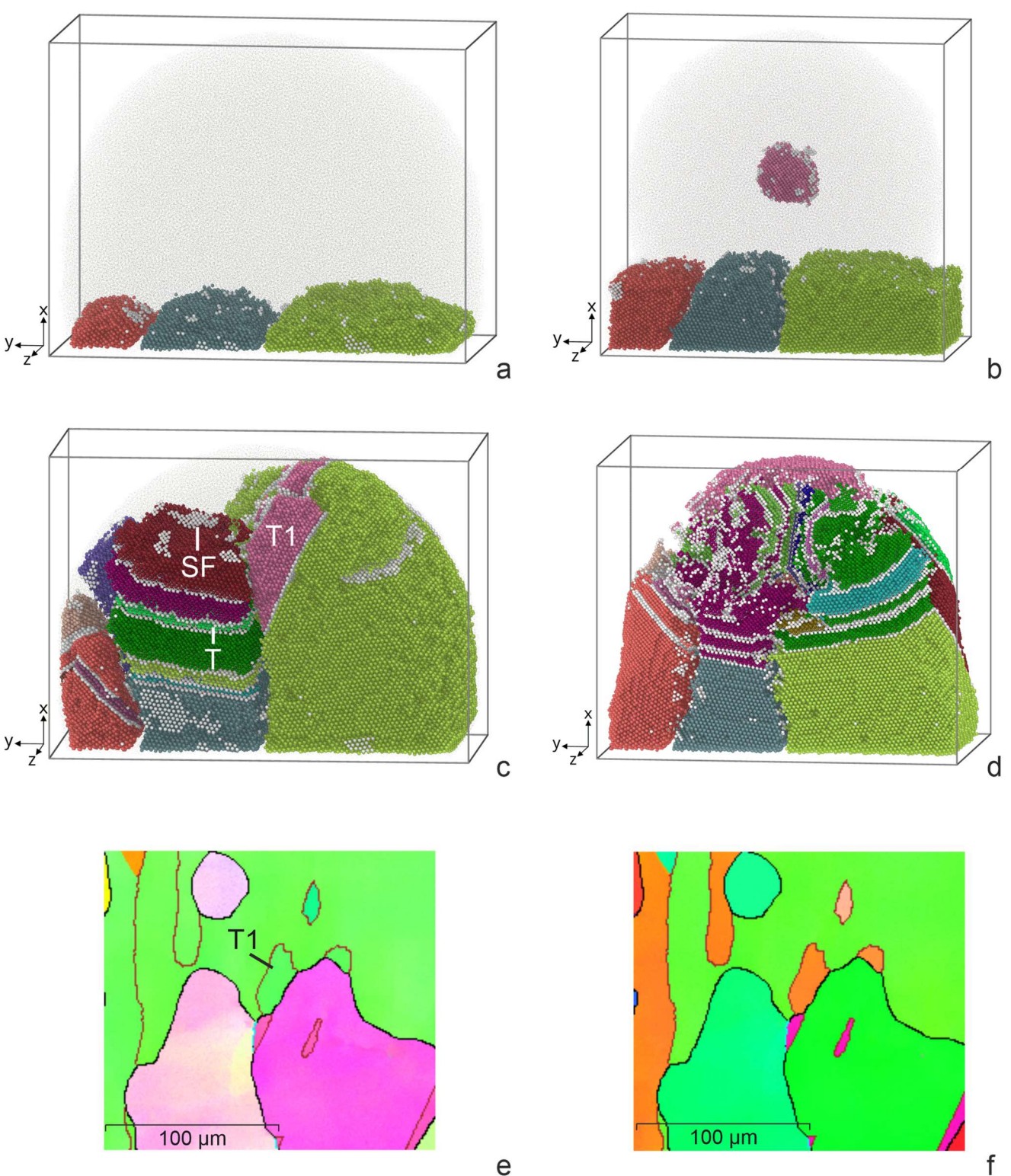

**Figure 7.** The structural changes during the interaction of a drop of melted aluminum bronze with the crystallized base (**a**,**b**) and subsequent crystallization (**c**,**d**) compared to orientation maps relative to the X (**e**) and Y (**f**) axes.

During crystallization, the largest number of twins was observed in the central grain, whose growth had to be consistent with the growth of neighboring grains. Note that in the case of the presence of an additional center of crystallization (grain 4) in the drop, the

number of SF and twins in the contact zone of four grains was the maximum. We see a similar picture in the crystallization grains of the deposited layer.

*5.2. Surfacing on a Deformed Base*

Let us consider the results of the simulation of the interaction of a melt drop with a deformed base at different drop temperatures. The initial structure of the deformed three grains is shown in Figure 6. Figure 8 shows the nature of the interaction of a drop with a temperature of 1380 K with a three-grain base. The disappearance of the fraction of stacking faults formed during deformation was observed. Due to the low temperature of the melt, some stacking faults remained while the drop propagated to a limited part of the surface. This caused the grains to grow unevenly. Of the three grains, only two formed elongated grains in the direction of the temperature gradient. In the process of crystallization, the formation of stacking faults and twins (SF and T in Figure 8b) was observed, the latter being predominant. During crystallization, the boundaries between grains changed their orientation. The predominant growth was observed in the central grain.

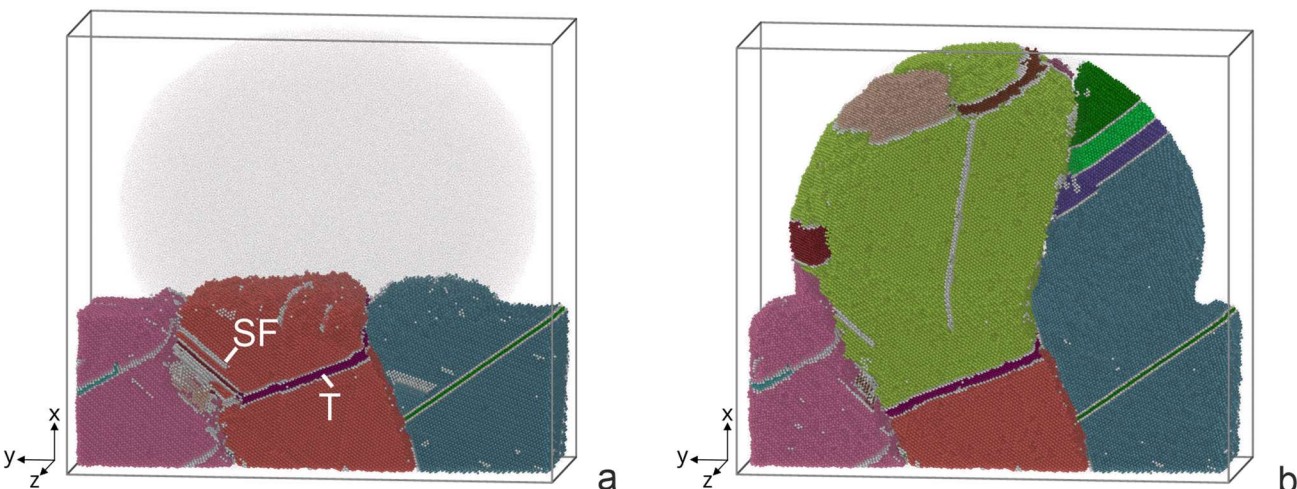

**Figure 8.** The structural changes during the interaction of a drop of melted aluminum bronze with the deformed base (**a**) and subsequent crystallization (**b**) (the drop temperature being 1380 K).

An increase in the heating temperature of the drop to 1500 K promoted an increase in the contact area of the drop with the surface and a deeper heating of the base. The temperature became sufficient for the partial melting of the surface and the disappearance of the upper SF in individual grains (Figure 9a). The formation of SFs and twins and the growth of initial grains were observed during crystallization (Figure 9b). Twins and SFs, as a rule, have orientations similar to their orientations in the original grains. When grains grow, the boundaries also deviate from their initial orientation. The growth of grains in this case was uniform according to the grain structure of the base.

Intensive spreading of the drop over the base surface and significant melting of the base were observed during the interaction of a melt drop with a temperature of 2000 K. A single SF remained near the substrate (Figure 10a). During cooling, twins were formed only at the end of crystallization, and the amount of SF was small. Deviations of grain boundary planes from the initial orientation were observed.

Thus, the results of the numerical experiment revealed that, depending on the ratio of the heat of the melt and the volume of the base, as a heat-absorbing element, different crystallization structures could be realized. The more perfect the base material, the less SF and growth twins in the crystallization structure. The appearance of the observed defects at the final stage of crystallization is apparently due to the uncoordinated growth of adjacent grains. Twinning is a mechanism for changing the orientation of grains during their growth. The main direction of growth corresponds to the maximum temperature gradient. The

growth rate is influenced by the amount of melt inflow and the absence of interference from neighboring grains. During crystallization, the influence of crystallization centers inside the melt cannot be ruled out, along with crystallization from the base grains, from which the most intensive heat removal is carried out. The results of crystallization simulation via the MD method are in qualitative agreement with the experimentally observed character of grain growth.

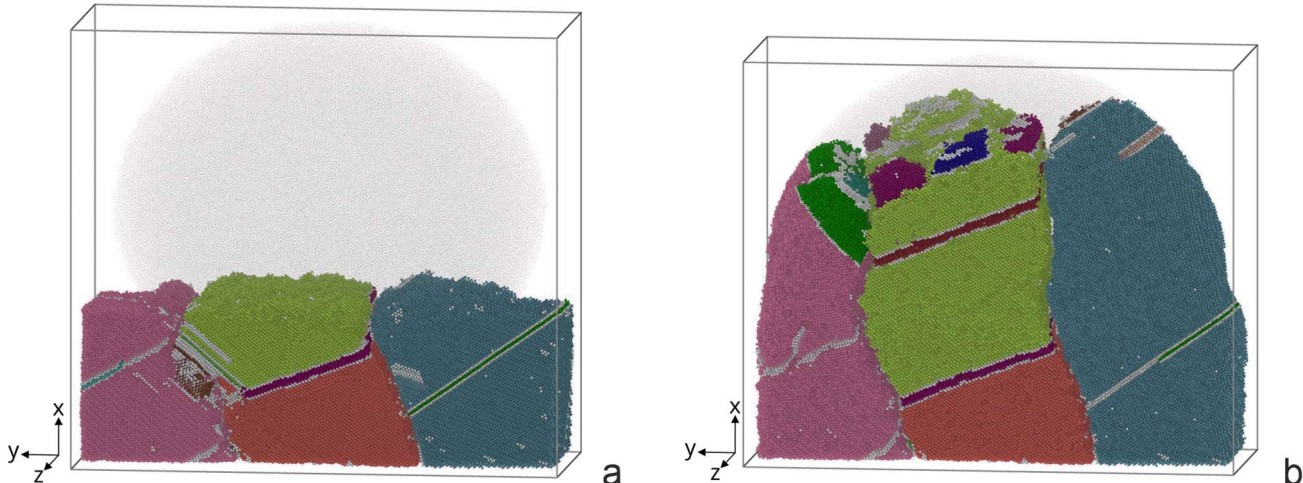

**Figure 9.** The penetration by a drop of melted aluminum bronze of the deformed base (**a**) and its subsequent crystallization (**b**) (the drop temperature being 1500 K).

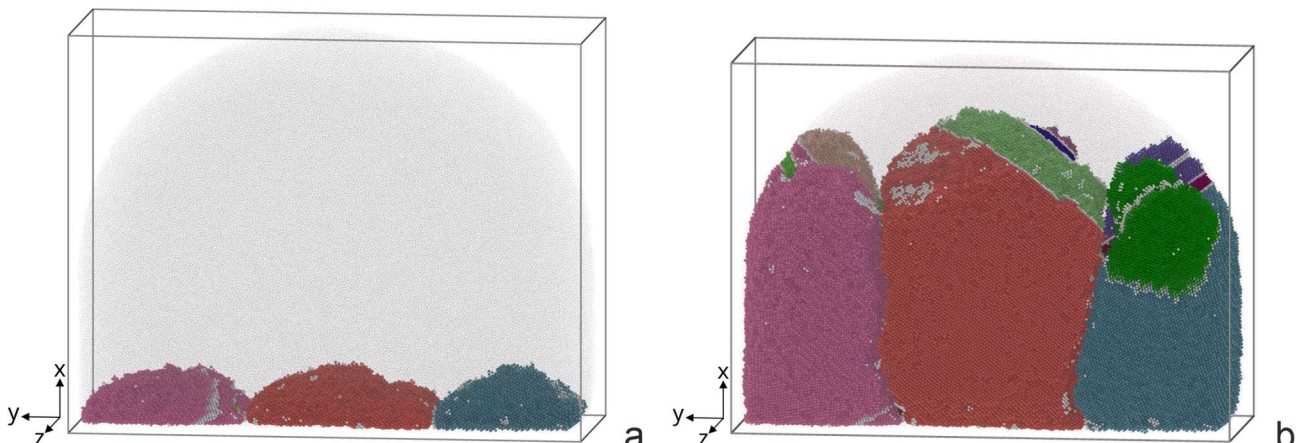

**Figure 10.** The melting of the deformed base (**a**) with the melting drop with a temperature of 2000 and subsequent crystallization of the drop (**b**).

## 6. Discussion

One of the main factors affecting the structure of a material during crystallization in the process of surfacing metallic materials is the amount of heat input and the rate of dissipation of this heat. What is important is the volume of the crystallizing melt and structural factors that determine the number and role of crystallization centers. Let us note some works being carried out in this direction. Liu Y. et al. [15] using the Ti-22Al-25Nb alloy showed that a change in the deposition rate noticeably affects the characteristics of the microstructure (grain size, texture, and other parameters). By controlling the surfacing rate, it is possible to obtain the necessary structural parameters that determine the optimal mechanical properties.

Filippov A. et al. [23] carried out studies on aluminum bronze surfacing using a varying heat input. Technological regimes were established under which the grain size

increases and the transition from an equiaxed to columnar grain structure occurs. It is shown that grain growth is inhibited by fast heat removal and low heat input. The considered experiments explain the main patterns of structural changes observed by us during surfacing on a deformed base in various parts of the remelting zone.

The surfacing technology used involves filling the layer area with parallel lines in a zigzag or sequential pattern of their overlay. At the same time, adjacent sections are subjected to reheating and (or) remelting to different overheating temperatures above the melting point. This circumstance was reflected in the simulation modes through setting different temperatures of the melt drop. At a lower temperature of 1380 K, in addition to a moderate degree of melting of the base, there was a limitation to the spreading of the melt over the surface of the base. This agrees with the data on the variation in heat input. The authors of [15,23] note the presence of areas of poor penetration with increased porosity. Increasing the temperature avoids these undesirable effects. Using temperatures of 1500 K and 2000 K in modeling, good interaction between the melt and the base was achieved. An increase in temperature in this interval affects the inheritance of faults from the base through the depth of remelting. This circumstance must be taken into account when choosing the heat temperature. The design of the experiment in this work does not allow a revelation of these effects in pure form, since the repeated heating and cooling cycle during the application of each subsequent layer makes its own impact.

Regularities of crystallographic growth during surfacing are considered. For this, the calculation model was based on the crystallographic parameters of a group of grains and the crystallization process was simulated. Grain growth was assumed to follow the direction of the maximum heat gradient. The upper crystallizing grains served as an indicator of the process and were compared with the structure of the upper grains on the orientation map of the selected area. The simulation showed a good agreement with the experiment regarding the sites of twin nucleation. It should be noted here that at the moment, the method for determining the boundary plane requires the construction of a 3D orientation map, which was not carried out in this work. Therefore, the assignment of the boundary plane during simulation may differ from that observed experimentally in the selected area.

Summing up the discussion of the results, we note the good possibilities of the MD method for modeling crystallization processes, taking into account the spatial and temporal limitations imposed by computing power.

## 7. Conclusions

Experimental studies of the surfacing of aluminum bronze on a deformed base showed a difference in the structure in each selected layer (zone). The revealed structural difference is explained from the point of view of the local heat input's heterogeneity, which includes repeated remelting to different overheating temperatures and a thermal heating and cooling cycle with different parameters for different areas.

The revealed experimental regularities were used as the basis for the creation of simulated MD objects and modes for conducting a numerical experiment. The formation of a crystallization structure during the surfacing of aluminum bronze on a deformed base was traced at temperatures of 1380 K, 1500 K, and 2000 K (different heat inputs with the same heat removal scheme). The deformation structure was created by compressing the workpiece in a numerical experiment using the MD method. A simulation of a melt drop on an undeformed base was carried out. Two variants of crystallization were considered: only on a base and with a crystallization center inside a melt drop.

The numerical experiment made it possible to establish that the amount of heat input affects the degree of melting, leading to the destruction of the faulted structure of the deformed base. In this case, subsequent crystallization is accompanied by a difference in the defectiveness of the growing crystals. The fewer defects there were in the base, which plays the role of a seed, the fewer defects there were in the growing crystals. It has been established that crystal growth is accompanied by a change in the orientation of

microregions. The considered case of modeling showed that in this case, boundaries of general and special types are formed. Statistical consideration of misorientation boundaries and the literature data indicate a low density of special boundaries in the deposited layers. We experimentally found that low-angle boundaries with a misorientation angle of up to 15 degrees are more favorable for the consistent growth of neighboring grains. The simulation showed that during the growth of grains, a change in the direction of grain growth along the line of maximum heat removal is observed. This manifests during surfacing. The slope of the columnar grains changes in the direction of the electron beam along the weld. The results of the simulation qualitatively agree with the experimental data on the localization of the sites of new grain formation during crystallization.

Analysis of the microstructure showed that the use of intermediate volumetric deformation does not significantly affect the improvement of the substructure. It is promising to reduce the heat input, which reduces the possibility of the formation of columnar grains.

Thus, carrying out a natural and numerical experiment made it possible to reveal the effect of thermal input on the features of a local change in structure during heating in the process of surfacing new layers of aluminum bronze on a deformed base. The agreement between the results of the MD simulation and experimental results and the prospects for a numerical experiment are shown.

**Author Contributions:** Conceptualization, D.V.L. and A.Y.N.; writing—original draft preparation, D.V.L. and A.Y.N.; software, A.Y.N. and A.A.B.; writing—review and editing, A.A.B.; experiment D.V.L. and O.S.N. All authors have read and agreed to the published version of the manuscript.

**Funding:** This research was funded by the Russian Science Foundation (project 20-72-10184; link to information about the project: https://rscf.ru/en/project/20-72-10184/ (accessed on 20 May 2023).

**Data Availability Statement:** The data presented in this study are available upon request from the corresponding author. The data are not publicly available due to privacy.

**Conflicts of Interest:** The authors declare no conflict of interest.

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
