# Peer review of "Aluminum Bronze Crystallization on Deformed Base during Electron Beam Additive Manufacturing"

_metals, doi:10.3390/met13061012_

Round 1

Reviewer 1 Report

In this paper, aluminum bronze was surfacing onto a deformed base and the structure and its characteristics in the surfacing and heat-affected zones were studied by the EBSD method. The following modifications are required before it is considered for publication.

(1) special type boundaries (CSL), this abbreviation is not appropriate.

(2) What is the difference between the grains in Fig.5c, please marked in the Fig.5c.

(3) What does the atomic color in Fig.6 mean.

(4) The change in color of the atoms in Fig.7, Fig.8, Fig.9 and Fig.10 confused the reader.

(5) What is the significance of the authors performing this simulation. What is the relationship between molecular dynamics simulations and EBSD results.

Minor editing of English language required

Author Response

Thank you for attentive reading and valuable comments. They have allowed us to improve the manuscript a lot. Please find the replies below. We have also made some minor changes in the manuscript according to your comments and suggestions. All the modifications were marked up using the “Track Changes” function by MS Word in the revised manuscript. The point-to-point responses to the reviewer’s comments are listed as following:

1. Special type boundaries (CSL), this abbreviation is not appropriate

The abbreviation has been changed to STB.

2. What is the difference between the grains in Fig.5c, please marked in the Fig.5c.

Difference between grains is the orientation of the lattice. We add schematic orientation of lattice for each grain on fig 5c.

3. What does the atomic color in Fig.6 mean.

The colors show the atoms belonging, according to PTM, to different grains. We have added one explanation below Figure 5 for all further structures in Figs. 6-10. In text we add information about atoms in HCP lattice.

4. The change in color of the atoms in Fig.7, Fig.8, Fig.9 and Fig.10 confused the reader.

Figures 7 and 8-10 correspond to different orientations of the substrate grains. In the first case, the grain colors correspond to the substrate in Fig. 5c, in the second – Fig. 6.

5. What is the significance of the authors performing this simulation. What is the relationship between molecular dynamics simulations and EBSD results.

The correspondence between modeling and EBSD results is repeatedly discussed throughout the text of the work. The generalized result is given in the last paragraph of the conclusion.

Reviewer 2 Report

The abstract should be written better and needs major revisions. The purpose of research and innovation should be clearly stated. Also, the performed tests should be presented first, and then the results should be presented quantitatively and qualitatively. The choice of keywords is inappropriate. For example, additive manufacturing is not mentioned in the abstract but is one of the keywords.  Abbreviations are permitted if they are fully defined for the first mention (EBSD). The article needs general writing and grammar editing.

The first paragraph of the introduction should be amended, and the said items are valid in the field of metal printing. The introduction is written very briefly, and at the end, a suitable summary of the importance of the present issue is not provided. SLM and molecular dynamic simulation need to reform and deepen the introduction. Use the following resources to complete this section.  Molecular dynamics simulation and experimental study of tin growth in SAC lead-free microsolder joints under thermo-mechanical-electrical coupling. Effect of welding thermal treatment on the microstructure and mechanical properties of nickel-based superalloy fabricated by selective laser melting. Investigation of welding crack in micro laser welded NiTiNb shape memory alloy and Ti6Al4V alloy dissimilar metals joints.

It is recommended to provide all printing parameters in one table. How to choose them should also be mentioned.

How have the numerical and simulation results been verified? Also, how has the used model been verified? Is the reproducibility of the results checked?

The results section is well organized and categorized. But some parts report the results, which require corrections and deepening the analysis and discussion. Use the suggested resources to deepen the discussion. Effects of post-weld heat treatment on the microstructure and mechanical properties of laser-welded NiTi/304SS joint with Ni filler and Microstructural origin and control mechanism of the mixed grain structure in Ni-based superalloys.

It is suggested that the results be presented quantitatively and the examined parameters in a table or a chart.

The scale bar and error bar is missing for some images and results.  Use the recommended sources to analyze the results. It is suggested to modify the conclusion section as well as the abstract.

No comment.

Author Response

Thank you for attentive reading and valuable comments. They have allowed us to improve the manuscript a lot. Please find the replies below. We have also made some minor changes in the manuscript according to your comments and suggestions. All the modifications were marked up using the “Track Changes” function by MS Word in the revised manuscript. The point-to-point responses to the reviewer’s comments are listed as following:

1.The abstract should be written better and needs major revisions. The purpose of research and innovation should be clearly stated. Also, the performed tests should be presented first, and then the results should be presented quantitatively and qualitatively. The choice of keywords is inappropriate. For example, additive manufacturing is not mentioned in the abstract but is one of the keywords. Abbreviations are permitted if they are fully defined for the first mention (EBSD). The article needs general writing and grammar editing.

Edits have been made to the abstract.

2.The first paragraph of the introduction should be amended, and the said items are valid in the field of metal printing. The introduction is written very briefly, and at the end, a suitable summary of the importance of the present issue is not provided. SLM and molecular dynamic simulation need to reform and deepen the introduction. Use the following resources to complete this section.  Molecular dynamics simulation and experimental study of tin growth in SAC lead-free microsolder joints under thermo-mechanical-electrical coupling. Effect of welding thermal treatment on the microstructure and mechanical properties of nickel-based superalloy fabricated by selective laser melting. Investigation of welding crack in micro laser welded NiTiNb shape memory alloy and Ti6Al4V alloy dissimilar metals joints.

Changes have been made to the introduction.

3. It is recommended to provide all printing parameters in one table. How to choose them should also be mentioned.

A table describing the mode of operation of the installation has been added to the text of the manuscript. Modes were used, the optimal choice of which was made on the basis of previously performed works [1]. The corresponding phrase has been added to the text of the manuscript.

Filippov, A.; Shamarin, N.; Moskvichev, E.; Savchenko, N.; Kolubaev, E.; Khoroshko, E.; Tarasov, S. Heat Input Effect on Microstructure and Mechanical Properties of Electron Beam Additive Manufactured (EBAM) Cu-7.5wt.%Al Bronze. Materials (Basel). 2021, 14, 6948, doi:10.3390/ma14226948.

4. How have the numerical and simulation results been verified? Also, how has the used model been verified? Is the reproducibility of the results checked?

The model is based on the proven potential of interatomic interaction, which correctly describes the processes of melting and crystallization of Al-Cu alloys. The proven open source software package LAMMPS was used to integrate the equations of atomic motion. 5 calculations were carried out with varying initial conditions (distribution of velocities of atoms). The calculation results give similar final atomic structures of the samples.

5. The results section is well organized and categorized. But some parts report the results, which require corrections and deepening the analysis and discussion. Use the suggested resources to deepen the discussion. Effects of post-weld heat treatment on the microstructure and mechanical properties of laser-welded NiTi/304SS joint with Ni filler and Microstructural origin and control mechanism of the mixed grain structure in Ni-based superalloys.

These works are used in the introduction. In the discussion, we decided to focus only on aluminum bronze alloys.

6. It is suggested that the results be presented quantitatively and the examined parameters in a table or a chart.

Within the framework of the work under consideration, all the necessary quantitative results are presented in Figs. 4b, as well as orientation maps and grain boundary maps. The discussed crystallographic parameters are quantitative parameters.

7. The scale bar and error bar is missing for some images and results.  Use the recommended sources to analyze the results.

We added error bar to the chart. Added scale to Fig. 7. The accuracy of the grating orientation angles determined by the EBSD method is 2 degrees.

8. It is suggested to modify the conclusion section as well as the abstract.

The abstract and conclusion have been amended.

Reviewer 3 Report

The paper entitled “Aluminum bronze crystallization on deformed base during Electron Beam Additive Manufacturing" presents the results of a study on the heterogeneity of the formation of the structure due to heat input in EBAM. From my point of view, the topic is of great interest and the quality is good. But in general:

·        The abstract provides a general idea of the research conducted, but it could be improved by providing a clearer and more concise summary of the main findings and conclusions. Additionally, it would be helpful to include a brief statement about the practical applications or implications of the research. Include too many keywords

·        The introduction seems to establish somewhat the objective of the article but not the novelty.

·        In the introduction to include the application of finite element modelling in direct energy deposition processes such as WAAM, I believe it can add a focus to the application of this type of work to other processes. As examples:

o   https://doi.org/10.3390/met11050678

o   https://doi.org/10.1016/j.commatsci.2011.06.023

·        From Figure 3 onwards all figures are in boxes which is not visually appealing.ç

·        Please include more macroscopic photos of the test samples after the surfacing process. Specifically, I would like to see a figure that show the test samples in their macroscopic state after the surfacing process.

·        The conclusion effectively summarizes the key findings of the study. However, it could be improved by adding some context about the significance of the findings and their potential applications.

Author Response

Thank you for attentive reading and valuable comments. They have allowed us to improve the manuscript a lot. Please find the replies below. We have also made some minor changes in the manuscript according to your comments and suggestions. All the modifications were marked up using the “Track Changes” function by MS Word in the revised manuscript. The point-to-point responses to the reviewer’s comments are listed as following:

1. The abstract provides a general idea of the research conducted, but it could be improved by providing a clearer and more concise summary of the main findings and conclusions. Additionally, it would be helpful to include a brief statement about the practical applications or implications of the research. Include too many keywords

Edits have been made to the abstract. The list of keywords has been shortened.

2. The introduction seems to establish somewhat the objective of the article but not the novelty.

Significant changes have been made to the introduction. The relevance of the work is more clearly indicated.

3. In the introduction to include the application of finite element modelling in direct energy deposition processes such as WAAM, I believe it can add a focus to the application of this type of work to other processes. As examples: https://doi.org/10.3390/met11050678 , https://doi.org/10.1016/j.commatsci.2011.06.023

Examples of using FEM to model directed energy deposition processes have been added to the introduction.

4. From Figure 3 onwards all figures are in boxes which is not visually appealing.

The Figures have been edited.

5. Please include more macroscopic photos of the test samples after the surfacing process. Specifically, I would like to see a figure that show the test samples in their macroscopic state after the surfacing process.

In fig. 1 added photo of the surface of the sample after surfacing.

6. The conclusion effectively summarizes the key findings of the study. However, it could be improved by adding some context about the significance of the findings and their potential applications.

Analysis of the microstructure showed that the use of volumetric intermediate deformation does not significantly affect the improvement of the substructure. It is promising to reduce the heat input, which reduces the likelihood of the formation of columnar grains. Appropriate text added in the conclusion.